# Using R Shiny to develop a dashboard using IPEDS, U.S. Census, and bureau of labor statistics data

Mark A. Perkins⊙*◉, Jonathan W. Carrier◉

College of Education, University of Wyoming, Laramie, Wyoming, United States of America

◉ These authors contributed equally to this work.
* mperki17@uwyo.edu

**Data Availability Statement:** Data are downloadable from the application and the programming code is provided as well.

**Funding:** The authors received no specific funding for this work.

## Abstract

This paper describes the development of an RStudio (now known as Posit) dashboard derived from the Integrated Postsecondary Educational Data System, the United States Census Bureau, and the Bureau of Labor Statistics and provides the user with institutional, community, and career information of IPEDS reporting higher education institutions in the United States and its territories. With this dashboard, users can select and learn about institutions, explore enrollment trends and demographics, compare outcomes, and correlate community and institutional variables. Users can also link degrees to career projections and wages. This paper explains how the dashboard was developed with examples of R programming language.

## Introduction

R is a statistical programming and graphics software that allows for powerful analytics and data visualizations, including the creation of interactive dashboards [1,2]. Within the public domain, there are several available datstsets from which researchers, data scientists, and other analysts can draw to conduct analyses. For example, three such databases in the United States are the United States Census Bureau (USB) [3], the Integrated Postsecondary Data System (IPEDS) from the National Center of Educational Statistics (NCES) [4], and the Bureau of Labor Statistics (BLS) [5]. Additional public datasets are available to such fields as healthcare, transportation and law enforcement. In addition, publicly available datasets are present in several nations. The use of these large public data systems can be challenging. Several open-source internet sites provide information on how to generate and program data analytics tools, but few resources exist that show how to combine large public data sets in a user-friendly dashboard. Here, we present a model Shiny application that uses publicly available data to provide interested analysts and researchers with instructions on how query, munge, manipulate, and program data to develop a dashboard tool.

The purpose of this paper is to demonstrate how to query and munge publicly available IPEDS, United States Census Bureau (USCB) and Bureau of Labor Statistics (BLS) data to then program and develop a Shiny application. The resulting dashboard provide users with

**Competing interests:** The authors have declared that no competing interests exist.

information about enrollment, graduation rates, demographic information, correlations between college and community factors, as well as employment and income projections by classification of instructional programs (CIP) code and programs by institution. This paper will teach data scientists, data analysts or other professionals how to extract from data sets and program similar applications. In so doing, the reader will learn about the datasets themselves, and they will learn about the R packages and programming techniques. This paper presents a dashboard using variables of interest to higher education professionals, but thestrategies and programming code presented here will also be useful with other public data sets in such fields as transportation, healthcare, and law enforcement.

These datasets contain thousands of variables. IPEDS has 250 variables [4], the USCB over 18,000 [3] and BLS has several datasets containing tens of thousands of variables (we could find no exact number) [5]. For this demonstration, we choose variables related to demographics, educational outcomes and labor outcomes. However, other programmers may choose to use any number of variables from these datasets depending on the goals of their project. No matter what variables are chosen, it is imperative that the programmer studies the variable sources, code books, and logic. This is the case whether they choose these demonstrated datasets, or other datasets. It is our goal to use the variables in this paper to demonstrate basic coding and dashboard design skills that could be applied to any number of variables from any number of data sources.

First, we present an overview of the databases used for the study including where to locate these databases, and if relevant, how to access them directly through R statistical software using the *tidycensus* and *ipeds* packages [1,2]. Next, we demonstrate how the developer can use programming language in the *tidyverse* package to consolidate and set-up their data for various interactive graphs [3]. Further demonstrations will illustrate the coding structure and logic to generate *ggplotly* interactive tables and graphs with dynamic and interactive input controls [4,5] to publish on an Shiny application. We also review the dashboard tab by tab and explain how each interface was developed. The application dashboard can be found here: https://ipedsdash.shinyapps.io/demonstration/ and the code used query and munge the data as well as the code to generate the web application can be found here: https://rpubs.com/IPEDS.

## Background

This section will provide an overview of the datasets used for this demonstration. First, it will go into thebackground and public availability of IPEDS, BLS, and USCB. These are just three examples of publicly available datasets that are accessible by anyone with an internet connection and an interest in studying them. Public datasets are available from many sources, nations, and organizations across a variety of content areas or topics.

### Integrated postsecondary data system

The integrated postsecondary data system (IPEDS) has been collecting data since the 1980s [6]. Currently, IPEDS collects data on over 7, 000 institutions on over 250 variables, and IPEDs data in one form or another is available from 1980 to the present, though methods of data collection have changed over time [7]. In addition, NCES and IPEDS offer different ways to obtain IPEDS data. Most of the data are available through the College Navigator which allows search and filter systems. Data is also available through the College and Career Tables library. The IPEDs datacenter allows users to create, compare, and export data as far back as the 1980s to compare institutions, look at trends, and develop benchmarking data [7]. Interested parties may also download all the IPEDS data on Microsoft Access files from the IPEDS directly, or using the R package *ipeds* [1,8].

Few peer reviewed studies have used IPEDS data for institutional analyses. Some literature discusses the use of IPEDS data to help institutions with benchmarking, or making comparisons with other institutions [9–11]. Other research used IPEDS data to determine true college costs [12], to examine cultural and racial equity in enrollment and outcomes [13,14], to examine productivity or to develop systems to measure productivity of higher education institutions [15], to examine graduation rates [16], examine completion and credentials by specific demographics and categories [17], and to examine the effects of online programs on state appropriations [18].

No peer-reviewed scholarly literature was found that combined IPEDs data with U.S. Census and Bureau of Labor Statistics data in publicly available dashboards. A search of non-peer-reviewed sources on Google also yielded no findings of such dashboards or search tools outside of what NCES provides individually in IPEDS.

## Bureau of labor statistics

The Bureau of Labor Statistics (BLS) was founded in 1884 under the Department of the Interior, and became independent from 1888 to 1903 when it was housed under the Department of Commerce and Labor. In 1913, it found its current home under the Department of Labor [19]. According to their website (https://www.bls.gov/bls/infohome.htm), since their founding, "The Bureau of Labor Statistics measures labor market activity, working conditions, price changes, and productivity in the U.S. economy to support public and private decision making" [20]. The BLS values just reporting the facts, complete transparency, confidentiality, gold-standard data, customer input, customer service, and innovation and serves a wide variety of stakeholders including decision makers, business leaders, consumers, economists, financial investors, jobseekers, the media, policy makers, educators, and researchers [20].

The Bureau of Labor Statistics (BLS) is housed under the department of labor [19] and collects wage, occupation, employment, and unemployment data [21]. These data may be of particular interest to higher education institutions as they examine the employment and wage outlook for their degrees and programs. Some literature links postsecondary credentials with labor and wage data. Crellin et al. [22] examined earnings and levels of education by state, finding that achieving the 60% educational attainment goal of the United States (at that time) may help improve economic outcomes.

## United states census bureau

A The United States Census Bureau (USB) collects geocoded data, or data coded and related to geographical locations, every decade on a vast number of demographic and economic metrics [23]. A detailed search in EBSCO, Web of Science, and related databases found little research linking higher education outcomes to specific USCB metrics. One study by Crellin et al. [22] examined these data in relation to college attainment goals, in conjunction with the U.S. Higher Education Management Systems (MCHEMS), and the Center for Law and Social Policy (CLASP).

## These data sets combined

Little peer reviewed literature demonstrates the use of publicly available data outside of benchmarking. Crellin et al. [22] found a positive correlation between college degree attainment and household income. Gonzáles Canché [23], wrote extensively on strategic ways to combine USCB and IPEDS datasets. In addition, some research used these data to examine the effects of local factors on neighboring tuition prices, the effects of non-resident students on tuition costs, and using geographical network analysis, a quantitative method using geographical

imaging and geocoded data, to examine student migration [24–27]. No research was found that links multiple public data systems to dashboards. This project sought to demonstrate how to develop a dashboard that integrates multiple public data systems.

## Methodology

This study was approved as exempt by the institution's review board. The data in this study and dashboard are all publicly available. This section provides an overview of the R packages used and then moves to data collection and then to programming and analyses of the dashboard application.

### R packages

S1 Table gives a list of the R package libraries that were used to generate this dashboard. These libraries handle everything from data querying and munging to generating plots and tables and developing the actual Shiny application [1–5,28–35].

### Querying the data

This section provides examples of the coding strategies used to query and munge the data. This link provides the programming code for all the data querying and munging: https://rpubs.com/IPEDS/Data_Pull. These examples present a general pattern of all the code required to gather the data for each of the interfaces of the dashboard. RMarkdown was used to query and munge the data in preparation for the dashboard tables, graphs, and interfaces [33]. The presented dashboard used data from three primary sources: (a) IPEDS, (b) USCB, and (c) BLS and required extraction of specific elements from specific tables of each of the data sources and then a joining of the data. In general, USCB data and BLS data were joined with IPEDS data using the county identifier in each dataset. County identifiers are unique numeric codes used universally across federal data systems, which means that county attributes were linked with institutions that resided in those counties.

When linking multiple data sets, we stress the importance of making sure that the dates and concepts align logically. Misalignment of data sets, even if joinable, renders invalid results, and worse, could lead to poor decisions given the bad data rendered.

**Gathering and joining IPEDS tables.** The IPEDS data system consists of multiple tables on a Microsoft Access file, each table consisting of numerous variables. Since these tables are separate, it is necessary to query individual variables from specific tables and then join those variables using the institution identifier, a unique code assigned to every IPEDS participating institution. The following code demonstrates how to request a specific dataset using the *ipeds* R package and then query specific tables. In this case, basic information about each institution is queried, then the table is reduced only to the variables of interest which include institution identifier (UNITID), the county code (COUNTYCD), state abbreviation (STABBR), institution name (INSTNM), institution alias (IALIAS), institution system (F1SYSNAM), longitude (LONGITUD), latitude (LATITUDE), locale (LOCALE), and Carnegie classification (C18SZSET) [1].

As shown in the code, using the *RODBC* package on R, the programmer links to the Access file provided by IPEDS. Next, using the same package, the programmer specifies which IPEDS data table to access. This code also limits the dataset only to public postsecondary institutions. Finally, using *tidyverse*, the programmer creates a subset of the data table to only include the variables of interest [1,34,39]. To describe in detail, the first sequence in this code, "IPEDSDatabase" constitutes the name of the database. By putting a "<-"after that name, we are telling R to name something "IPEDSDatabase". The sequence of the code after that is telling R to

connect to the IPEDS Access file with the command "obcDriverConnect()". This is followed by the type of driver to access and then the location of the Access file. Finally, a table is created called "institutioninformation" with the "sqlFetch" command and to pull from the IPEDS table named "HD2019". The code below that then tells R to limit the data table to a CONTROL of 1. The "%>% is known as a pipe and is a function of the *magrittr* package. This function takes the object on the left hand side and "pipes" it onto the first argument of the subsequent function. For example, *institutioninformation %>%select(UNITID)* is equivalent to *select(institutioninformation, UnitID)*. The pipe allows the code to be more readable than a series of nested functions.

```
IPEDSDatabase <- odbcDriverConnect("Driver = {Microsoft Access Driver
(*.mdb, *.accdb)};DBQ = C:/Users/mperki17/Documents/IPEDS201920.
accdb"
institutioninformation <- sqlFetch(IPEDSDatabase, "HD2019")
institutioninformation <- subset(institutioninformation, CONTROL = =
1)
institutioninformation %>%
select(UNITID, COUNTYCD, STABBR, INSTNM, IALIAS, F1SYSNAM, LONGITUD,
LATITUDE, LOCALE, C18SZSET)
```

Once the data are queried, it may be necessary to recode variables using *tidyverse*, a suite of R packages used to clean and munge data frames [3]. The following provides an example where the "LOCALE" variable, which consists of multiple levels, is re-coded into five general categories. As shown, the table is called "institutioninformation" and the variables are recoded using the *mutate()* and *case_when()* functions.

```
institutioninformation <- mutate(institutioninformation,
locale = case_when(LOCALE = = 11 ~ "City",
LOCALE = = 12 ~ "City",
LOCALE = = 13 ~ "City",
LOCALE = = 21 ~ "Suburb",
LOCALE = = 22 ~ "Suburb",
LOCALE = = 23 ~ "Suburb",
LOCALE = = 31 ~ "Town",
LOCALE = = 32 ~ "Town",
LOCALE = = 33 ~ "Town",
LOCALE = = 41 ~ "Rural",
LOCALE = = 42 ~ "Rural",
LOCALE = = 43 ~ "Rural",
LOCALE = = -3 ~ "Unknown"))
```

We also provide an example of how we recoded another variable called "C18SZSET", or the variable that determines institution type. IPEDS classifies institutions as two general categories of four-year (or bachelor degree institutions) and two-year (associate degree institutions), though they also have a category for institutions that only teach graduate students. These general categories are split into 19 specific categories ranging from "Two-year, very small" to "Not applicable". The following code illustrates how we reduced these to the categories of "two-year", "four-year", "Exclusively Grad.", and "Not Applicable". These 19 c were derived from the "valuesets18" table of the IPEDS Access file's data dictionary by filtering for the HD2018 table on the C18SZSET variable.

```
institutioninformation <- mutate(institutioninformation,
Type = case_when(C18SZSET = = 1 ~ "Two_Year",
C18SZSET = = 2 ~ "Two_Year",
C18SZSET = = 3 ~ "Two_Year",
C18SZSET = = 4 ~ "Two_Year",
C18SZSET = = 5 ~ "Two_Year",
C18SZSET = = 6 ~ "Four_Year",
```

```
C18SZSET = = 7 ~ "Four_Year",
C18SZSET = = 8 ~ "Four_Year",
C18SZSET = = 9 ~ "Four_Year",
C18SZSET = = 10 ~ "Four_Year",
C18SZSET = = 11 ~ "Four_Year",
C18SZSET = = 12 ~ "Four_Year",
C18SZSET = = 13 ~ "Four_Year",
C18SZSET = = 14 ~ "Four_Year",
C18SZSET = = 15 ~ "Four_Year",
C18SZSET = = 16 ~ "Four_Year",
C18SZSET = = 17 ~ "Four_Year",
C18SZSET = = 18 ~ "Exclusively_Grad",
C18SZSET = = -2 ~ "Not_Applicable"))
```

It is important to consider the objectives of a dashboard when conducting research with IPEDS or other data. For example, this particular dataset may not include every category of higher education institutions. For example, some institutions that are not public, only award less than two-year certificates, or other types of credentials may not be classified in the desired way by IPEDS or other data sources. Therefore, it is imperative of the researcher to choose datasets that meet their dashboard's objectives.

It may also be necessary to recode variable names so that they are easier to understand by the end-user of the final interface. The following example uses the "rename" function to change "INSTNM" to "Institution", thus changing the name of the column [3].

```
ipedsdashdata <- ipedsdashdata %>% rename("Institution" = INSTNM)
```

Building the dataset may require pulling several tables from various columns within the IPEDS database and then joining them using the institution identifier. The *valuesets19* and the *vartable19* tables in the IPEDS Access file (the 19 here signifies the IPEDS year, so 2020 IPEDS would have a 20) provide keys to all variables and codes. Using these tables will help determine which columns to pull from which tables, which then can be accessed using the *RODBC* package in the programming code as shown in the first code example. Once pulled, merge columns by using the join functions from the *dplyr* package [34]. The join functions allow for inner, outer, and left joins. The following code uses *left_join()* to combine the institution information dataset that was pulled from one table with the enrollment gender information that was pulled from another table. The left join means it will only join information that is relevant to the first (or left) table listed in the code, excluding anything in the other column that does not match. Here, a new table called "ipedsdashdata" will consist of a left join between a table called "institutioninformation" and a table called "enrollmentinformationgender", and it will link the two tables using the institution ID labeled "UNITID".

```
ipedsdashdata <- left_join(institutioninformation, enrollmentinforma-
tiongender, by = "UNITID")
```

The result is an unduplicated table with several columns of institution information that requires joining several IPEDS variables from several tables. Once completed, a single unduplicated table with all desired variables is generated for further analyses and programming of dashboard features.

## Gathering and joining census data

Combining institution information with USCB county data requires the use of *tidycensus* [2], a package that queries and builds datasets straight from the USCB directly into R. In order to use this package, the user must enter an application programming interface (API) key from the USCB. This key can be obtained from the USCB website by entering a name, email, and intended use of the data at this site: https://www.census.gov/data/developers.html. Sometimes websites change, so a simple Google search can get the user to the site [23]. The user also needs

to install the package on R and then load the library. Once they load the library, they need to enter their API key. Here, we first call the *tidycensus* library and then we load the library key.

```
install.packages('tidycensus')
library(tidycensus)
census_api_key("key_goes_here")
```

After entering the key, we recommend downloading the USCB variable dictionary and writing it as a.csv file. This file serves as a data dictionary to all of the metrics that are provided. The following code shows how to write the key as a database and then save it as a.csv file. In this case, the file is saved in the same folder as the RMarkdown file and is called "censuskey. csv". The user is able to open this in Excel and explore the different variables and how they are coded. This code shows how the variable key for the 2019 census data was generated and then written to a.csv file.

```
key <- load_variables(2019, "acs5", cache = TRUE)
write.csv(key, "censuskey.csv")
```

It is necessary to study and familiarize oneself with the USCB data. Each individual metric, or variable, comes with a code, and the USCB does not calculate proportions in the *tidycensus* dataset. Thus, the user is required to make these calculations directly in R. This takes time to do, but once familiar with the data, it becomes easier. The following example illustrates the code to calculate the proportion of the population with health insurance by county creating a table called "Health". This code shows the gathering of the variables using the *get_asc()* function. Then, with *tidycensus*, the summing of the variables and then the calculation of the proportion after the joining of the tables. This code also replaces NA values in cells with zeros using *replace_na()* so calculations are possible. Thus, this uses a combination of *tidycensus* and *tidyverse* [2,3,34].

This example pulls health insurance data by county. First, it creates a table called "Health-Tot" and then queries the county level data by category. For example the total number of white people with health insurance is "C27001B_007" from the *tidycensus* tables. It pulls all other categories then sums them. It then reduces the data table to GEOID, NAME, variable, and estimate using *select()*. It transposes it so that each category is its own column, replaces the null values with zeros, then sums the values into a now column so that the table shows total population with health insurance by county. It then joins with the population total and divides while renaming variables for clarity.

```
#Total with health
HealthTot <- get_acs(geography = "county", variables = c(Whitet =
"C27001A_007",
Blackt = "C27001B_007",
Nativet = "C27001C_007",
Asiant = "C27001D_007",
Pacifict = "C27001E_007",
Othert = "C27001F_007",
TwoMoret = "C27001G_007",
WhiteNott = "C27001H_007", Hispanict = "C27001I_007"))
HealthTot <- HealthTot %>% select(GEOID, NAME, variable, estimate)
HealthTot <- HealthTot %>% spread(variable, estimate)
HealthTot <- HealthTot %>% replace_na(
list(Whitet = 0,
Blackt = 0,
Nativet = 0,
Asiant = 0,
Pacifict = 0,
Othert = 0,
TwoMoret = 0,
WhiteNott = 0,
```

```
Hispanict = 0))
HealthTot <- HealthTot %>% mutate("HTot" = Whitet + Blackt + Nativet
+ Asiant + Pacifict + Othert + TwoMoret + WhiteNott + Hispanict)
#Join the health variables
Health <- left_join(HealthTot, HealthPro, by = "GEOID")
Health <- rename.variable(Health, "NAME.x", "County")
Health <- select(Health, GEOID, County, HPro, HTot)
Health <- Health %>% mutate("NoHealth" = HTot/HPro)
Health <- Health %>% mutate("WithHealth" = 1-NoHealth)
```

To obtain all the data required, several iterations like this may be necessary. Once all metrics are obtained, a final join of all the tables on county code will build an unduplicated USCB dataset that can then be duplicated with the master IPEDS dataset for a final table with all desired variables. The following code shows what this dashboard used. Each join was done individually to check each iteration. The final tables were also written into.csv files, but this is not always necessary.

```
#Final join for Census data
census <- left_join(employ, Health, by = "GEOID")%>%
rename.variable(census, "GEOID.x", "GEOID")%>%
left_join(income, by = "GEOID")%>%
left_join(TotalWhite, by = "GEOID")%>%
left_join(Veteran, by = "GEOID")%>%
left_join(HousePercent, by = "GEOID")%>%
left_join(Married, by = "GEOID")%>%
left_join(Education, by = "GEOID")%>%
left_join(Tribes, by = "GEOID")%>%
left_join(Parenting, by = "GEOID")%>%
left_join(Citizen, by = "GEOID")%>%
left_join(Renters, by = "GEOID")%>%
rename.variable(census, "County.x", "County")
write.csv(census, "census.csv")
#Join with IPEDS Database
ipedscensusdata <- left_join(ipedsdashdata, census, by = c("COUNTY-
CODE" = "GEOID"))
write.csv(ipedscensusdata, "ipedscensusdata.csv")
```

**Gathering and joining bureau of labor statistics data.** R has a Bureau of Labor Statistics (BLS) package called *blscrapeR* [35]. However, this package does not provide the BLS's cross-walk between institutional classification of instructional programs (CIP) codes and national standard occupational classification (SOC) code. Fortunately, the National Center for Educational Statistics (NCES) and the BLS provide a crosswalk between CIP and SOC codes, thus linking degree programs to occupations [36]. In addition, the BLS provides information about job projections and income [37]. To generate a table linking institution degree programs to job projections and income levels, R was used to download the CIP-SOC crosswalk.csv and the projections.txt, build a database of institutions' programs and CIP codes, then join them all into a single database. As shown in the code, several variable names were also changed from the data files. Note that *read.delim()* was used to read the.txt file and that file encoding was used when the.csv file was written. This function is a part of the *utils* package and is a part of the R core library [38]. The file was encoded to eliminate accents from labels, which can interfere with Shiny's ability to filter the data.

First, the CIP to SOC crosswalk file was pulled from the same folder as the RMarkdown file. Then, four variables were renamed using the *rename()* function.

```
CIPSOC <- read.csv("CIP_SOC.csv")
CIPSOC <- CIPSOC %>% rename("CIPCODE" = CIP2020Code)
CIPSOC <- CIPSOC %>% rename("Degree_Title" = CIP2020Title)
CIPSOC <- CIPSOC %>% rename("Career_Title" = SOC2018Title)
CIPSOC <- CIPSOC %>% rename("SOCCODE" = SOC2018Code)
```

Next, similar code was used with the wage dataset where 15 different variables were renamed. The *read.delim()* function was used to extract the data from a.txt file.

```
Wage <- read.delim("wage.txt")
Wage <- Wage %>% rename("SOCCODE" = OCC_CODE)
Wage <- Wage %>% rename("Total_Employed" = TOT_EMP)
Wage <- Wage %>% rename("Standard_Error" = EMP_PRSE)
Wage <- Wage %>% rename("Mean_Hourly" = H_MEAN)
Wage <- Wage %>% rename("Mean_Annual" = A_MEAN)
Wage <- Wage %>% rename("Meand_Standard_Error" = MEAN_PRSE)
Wage <- Wage %>% rename("Tenth_%ile_Hourly" = H_PCT10)
Wage <- Wage %>% rename("Twenty_fifth_%ile_Hourly" = H_PCT25)
Wage <- Wage %>% rename("Hourly_Median" = H_MEDIAN)
Wage <- Wage %>% rename("Seventy_Fifth_%ile_Hourly" = H_PCT75)
Wage <- Wage %>% rename("Ninetieth_%ile_Hourly" = H_PCT90)
Wage <- Wage %>% rename("Tenth_%ile_Annual" = A_PCT10)
Wage <- Wage %>% rename("Twenty_Fifth_%ile_Annual" = A_PCT25)
Wage <- Wage %>% rename("Median_Annual" = A_MEDIAN)
Wage <- Wage %>% rename("Seventy_Fifth_%ile_Annual" = A_PCT75)
Wage <- Wage %>% rename("Ninetieth_%ile_Annual" = A_PCT90)
```

Next, the wage data were merged with the CIP and SOC code data file and named "Wage". Then, the CIP codes of every institution in the United States were downloaded from the IPEDS dataset and written to a table called "instdegree". A third table was built from the "institution information" table that included identifiers, names, states, counties, locale, and type and named "instname." Variables were renamed in that table as "Institution Name", "Select a State," and "Community Type".

```
Wage <- left_join(CIPSOC, Wage, by = "SOCCODE")
instdegree <- sqlFetch(IPEDSDatabase, "C2018DEP")
instdegree <- instdegree %>% select(UNITID, CIPCODE, PTOTAL)
instdegree <- instdegree %>% rename("Total_Programs" = PTOTAL)
instname <- institutioninformation %>% select(UNITID, INSTNM, STABBR,
COUNTYCD, locale, Type)
instname <- instname %>% rename("Institution_Name" = INSTNM)
instname <- instname %>% rename("Select_a_State" = STABBR)
instname <- instname %>% rename("Community_Type" = locale)
```

The "instname" and "instdegree" tables were joined with the UNITID variable to create a "degree" table, which was then joined with the wage table by CIP code. A final table called "Degrees_and_Jobs" was generated from "degree" using the *select()* function and written to a. csv file with encoding. The encoded data rids the file of accents or markers that would interfere with R's ability to read and function with the data. The result is a data set that provides a link between programs and career projections and income by higher education institution. The table is named "Degrees_and_Jobs."

```
degree <- left_join(instname, instdegree, by = "UNITID")
degree <- left_join(degree, Wage, by = "CIPCODE")
Degrees_and_Jobs <- degree %>%
select(Institution_Name, Select_a_State, Degree_Title, Career_Title,
CIPCODE, SOCCODE, Total_Programs, Total_Employed, Mean_Hourly, Mean_-
Annual, Hourly_Median, Median_Annual)
write.csv(Degrees_and_Jobs, "Degrees_and_Jobs.csv", fileEncoding =
"UTF-8")
```

## Programming the dashboard

The combined IPEDS, USCB and BLS CIP/SOC data sets serve as platforms for the different elements of the dashboard. Querying and munging data, in our view, is the most laborious but

crucial element of any form of analyses, whether it be dashboard creation or inferential statistics. However, once the foundational data are obtained, one must not assume that the data munging has ended. As will be seen, it is often necessary to further customize datasets to meet the needs of the dashboard. To develop this dashboard, we first developed each tab on the dashboard on an RMarkdown file. This allowed us to test our programming code's functionality. After the elements were coded on RMarkdown, they were transferred to a Shiny application file, which required additional coding and logic before the actual dashboard was read to be published.

A Shiny dashboard is created holistically while it is also created in parts, making it a challenge to learn dashboard programming in R. When learning this code, it is important first to study the pattern of a Shiny dashboard. The first requirement is to load all necessary libraries. The next requirement is to load and label all the data. This dashboard loaded data from.csv files that were custom generated and written from RMarkdown. Each section will show how these files were generated and written, but it is important to know that these.csv files were saved in the same folder as the Shiny application file, thus making them easier to write. In addition, some datasets can be further customized as needed on the application file.

After the data are downloaded, the user interface (ui) is programmed. The ui includes the dashboard header, side bar menu items, and dashboard body content within each tab including code to link specific graphs and interfaces to specific tabs. After the interface is the server, which basically builds the graphics and interfaces for the users. This dashboard starts with all the user input controls with names linked to specific graphs. The following sections will include code for generating these input controls. Within the server are the outputs, which are the specific graphics of the dashboard.

Programming a dashboard on Shiny requires grit and patience as well as strong organization and documentation skills. *Shiny* dashboards are complex with several elements that link and communicate between the ui and the server and thus outputs. The easiest way to learn programming on R is to start with a simple application and then build it as it is expanded. For example, we might set up the ui and server for one simple application with one menu item on the sidebar. Once that works, we will build another one. Therefore, the application is never built in a linear fashion. All parts, from ui to the final line of code, grow simultaneously as the application is developed. This next section will go through the dashboard tab by tab and provide a conceptual overview of each's development as well as coding examples.

## Getting started with shiny

When programming a Shiny web application, the first thing to do is open a Shiny file. This is the space to program the application. This link provides the programming code for the demonstrated application: https://rpubs.com/IPEDS/dashboard. As shown, the first task is to load the necessary libraries to run the application and then to load the data. The data for this demonstrated application were generated in the *RMarkdown* file and saved as.csv files in the same folder that the Shiny application file. As shown in the linked program code, additional data munging was done in Shiny for the example used in this paper. Some users may choose to format everything prior to writing.csv files from *RMarkdown*. Others may choose to pull and munge their data in Shiny. We prefer to do munge our data in *RMarkdown* to avoid crowding ouShiny file, but it is really a matter of user preference.

**User interface and server.** Below these paragraphs is an excerpt of the programming code for the user interface tab. The first term in the demonstrated programming code, "ui", stands for user interface. Here Shiny is saying that the following programming will consist of the content that the end user of the dashboard will see. This is why it is followed by "<-". As shown, the next sections of the code list elements of the user interface in succession. The first element

is the skin of the dashboard, which is purple. Then it contains header elements. A main sidebar function, *dashboardSidebar()* commands Shiny to program the sidebar menu items that the user will see when going to the application. As shown, the first sidebar item is *menuItem()* which is beneath *sidebarMenu()*. This is the introduction of the dashboard, which will be the first page. The rest of the programming includes the arrangement of the page including the addition of a.png picture file in the interface as well as the *downloadButton()* features that allow the user to download data.

The link to the RPubs page gives the complete programming code for the welcome tab as well as the other tabs. The welcome tab is labeled as "tabName = 'intro'" under the *tabItem()* function, which is a subset of *tabItems()*. A careful studying of the programming language will show that each tab is programmed in the menu bar (i.e. "intro", "instmap", "histenroll", etc) and that each of those tab names are later linked to specific content of specific tabs. Thus, Shiny begins with a programming of user interface elements that then link to the mechanisms and content of the dashboard. The ui is therefore what the user sees. After the ui section of the dashboard is the server section which includes the outputs. These are the mechanisms working behind the things that the user sees in the ui. We think of the server functions as the gears and mechanisms behind the face of the clock. The server section usually begins with a "server <-"and includes the graphics of the dashboard that link to the ui elements. The server outputs are typically labeled as "output$" followed by the link to the ui. The sections of this paper covering each tab of the dashboard will describe their outputs. The below section of code shows how the ui was programmed. The rest of the code can be seen on the link. Whether programming ui or server outputs, it is important to keep track of parentheses and brackets as those accumulate and the logic within and between them matter.

```
ui <- dashboardPage(skin = "purple",
dashboardHeader(title = "IPEDS Dashboard [2019 Data)"),
dashboardSidebar(
  sidebarMenu(
    menuItem("Introduction", tabName = "intro", icon = icon("user")),
    menuItem("Institution Map", tabName = "instmap", icon = I con
    ("user")),
    menuItem("Historic Enrollment", tabName = "histenroll",
    icon = icon("user")),
    menuItem("Demographics", tabName = "demos", icon = icon("user")),
    menuItem("Graduation and Retention", tabName = "ccgraduation",
    icon = icon("user")),
    menuItem("Dynamic Scatterplot", tabName = "correlations",
    icon = icon("user")),
    menuItem("Correlation Coefficients", tabName = "matrix",
    icon = icon("user")),
    menuItem("Degrees and Careers", tabName = "jobs", icon = icon
    ("user")),
    menuItem("Degrees and Job Projections", tabName = "projections",
    icon = icon("user")),
    menuItem("Data Dictionary", tabName = "dictionary", icon = icon
    ("user")))),
dashboardBody(
  tabItems(
    tabItem(tabName = "intro",
    img(src = "image.png", height = 180, width = 320],
      h1("IPEDS Dashboard", align = "center"),
      h2("About this Dashboard"),
##Program the rest of your UI
)}
```

**Input controls and other features.** Shiny offers users a variety of features and widgets that allow the end user to explore the data, one of which consists of different input controls that allow the user to filter and slice the data. For example, a user may click on an input control and select a category to change the graph to only see that category. Shiny allows for several types of input controls, but this application uses two of them. The first, *selectInput()* allows the user to select from a dropdown list of options to choose one or more categories of a column variable. The second, *pickerInput()* allows the user to check one or more categories of a column variable.

On the "server <-" section of the RPubs link, the input controls programming is done in succession starting after the correlations tab was program and starts with the below programming code. In this paper, we explain how each of these were programmed while we go through each tab. However, the reader can refer to the link to see the geography of our programming language in the application. The point is that the input controls should be programmed on the server and linked to specific names in the ui. The example shows the first 10 lines of programming code in the sever section of the Shiny application file. The input control shown allows for a person to filter by state and colleges. We cover this later in the paper as we go through each tab of the application.

```
server<-function(input, output, session) {
df0<-eventReactive(input$stateInput, {
GradRate %>% filter(State %in% input$stateInput)
})
output$instInput<-renderUI({
selectInput("instInput", "Next Select One or More Colleges:", sort
(unique(df0()$Institution)), selected = "Casper College",
multiple = TRUE)
})
```

In addition, this demonstrated application also makes use of tooltips, which allow the user to hover their mouse over graphs or other data features and get additional information. It also has features that allow the user to filter with legends. Finally, there are widgets that allow the user to download data, which we use on the first tab of our ui (the code for which is on the RPubs link), and other features available in their literature [1]. The programming of these features are available on the RPubs link and examples are given in this paper when we cover each tab.

**Welcome tab.** The first page that pops up when users visit the dashboard is the welcome page. It was coded on the Shiny app file's interface using hypertext markup language (HTML) to include written text and pictures. After the *dashboardBody()* command, this section, as stated, begins with "tabItems(tabname = "intro",". Beneath that is a succession of HTML providing the user with visuals and specific text. The *img()* command tells Shiny to pull an image file named "image.png" into the dashboard and specifies its demensions and position. The next commands include the h1(), h2(), and h3() functions and tell R, what level of heading to use. The lower the number, the larger the font. Each h_() command is then followed by portions of text that are placed verbatim for the reader to use. This example reduces the quoted text, but tcomplete programming code can be found on the RPubs link. In addition, this section of text includes the option for the user to download the data in the dashboard using three downloadButton() commands, each linking to specific datasets described on this tab of the dashboard application. This occurs after a "fluidRow(box())" function. This generates a box in a row that adjusts to screen width. Each button is labeled as "Download Institution Data", "Download Degree and Job Data" and "Download Career Projection Data".

```
dashboardBody(
tabItems(
```

```
tabItem(tabName = "intro",
img(src = "image.png", height = 180, width = 320),
h1("IPEDS Dashboard", align = "center"),
h2("About this Dashboard"),
h3("This dashboard is still under development..."),
h2("Using the Dashboard"),
h3("The left bar...."),
h2("R Packages"),
h3("The development of this dashboard..."),
h2("Data Sets"),
h3("The buttons below..."),
h3(HTML("<p> If you would like to see the complete programminmg code,
the link to the RPubs page
<a href ='https://rpubs.com/IPEDS' is here.</p>")),
    fluidRow(
        box(width = 10,
        downloadButton("dataset", "Download Institution Data"),
        downloadButton("Degrees_and_Jobs", "Download Degree and Job
        Data"),
        downloadButton("Career_Projections", "Download Career Projec-
        tion Data"))),
h2("This dashboard will be updated..."))),
```

**Leaflet map.** The *leaflet* R package allows programmers to imbed customizable and searchable maps into a dashboard [32]. This tab of the dashboard includes a map of all public postsecondary institutions in the United States and its territories. Green circle markers indicate four-year degree institutions and purple markers indicate two-year degree institutions; a legend indicates such in the bottom right hand corner. The two-year and four-year institution types were coded from 18 different levels of institutions reported in the C18SZET table in the IPEDS database. IPEDS classifies institutions into 18 categories ranging from small two-year institutions to exclusively graduate institutions. For simplicity, we coded these into two categories, but other categories may be desirable by other programmers or users. The coding logic can be found on the linked site, which makes customization possible. The map also includes a Google search feature to zoom in on locations. There is a tooltip feature that allows the user to get graduation rates, fall enrollment numbers, cost off campus, and state in which the institution is housed.

This dashboard required a custom dataset from the master data file as shown on following code. The data were then written into a.csv file for use in the Shiny application. We place the. csv file into a "ShinyFiles" folder. This is where the Shiny application file resides. When we load the data into the application file, this file will be readily readable to the application.

```
mapdata <- select (ipedscensusdata, Institution, State, County, Type,
Community_Type, Longitude, Lattitude, Tot_Enrolled, TwoYGradRate150,
FourYGradRate150, Cost_Off_Campus)
write.csv(mapdata, "../Dashboard/ShinyFiles/mapdata.csv")
```

The map itself was coded as shown below under the server function. The resulting map allows the user to locate an institution of interest and quickly learn basic information about it. The following code begins with a "output$map < renderLeaflet({". This is telling Shiny to render the leaflet package using the map content as programmed in the ui (i.e. "menuItem("Institution Map", tabName = "instmap", icon = Icon("user"))"). The rest of the code details different options and features of the map including search options, marker options, and tooltip options. The *leaflet* site, provided in the table, gives good guidance on the choices one could make for their own project [32].

```
output$map <- renderLeaflet({
m <-
```

```
leaflet(mapdata) %>%
addTiles() %>%
addSearchOSM(options = searchOptions(zoom = 10, collapsed = TRUE,
hideMarkerOnCollapse = TRUE)) %>%
addCircleMarkers(group = "name", color = ~pal(mapdata$Type), fillOpa-
city = .8, lng = mapdata$Longitude, lat = mapdata$Lattitude, popup =
paste0("Name:", '\n', mapdata$Institution, '<br/>', "State:", '\n',
mapdata$State, <br/>', "Fall Enrollment:", '\n', comma(mapdata
$Tot_Enrolled, digits = 0),
'<br/>', "Cost off Campus:", '\n', paste0("$", comma(mapdata$Cost_-
Off_Campus, digits = 0)),
'<br/>', "Bachelor Grad Rate:", '\n', mapdata$FourYGradRate150%>%
paste0("%"),
'<br/>', "Associate/Cert Grad Rate:", '\n', mapdata$TwoYGradRate150%>
% paste0("%"))) %>%
addLegend("bottomright", pal = pal, values = ~mapdata$Type, title =
"College Type", opacity = 1) %>%
setView(lng = -98, lat = 38.87216, zoom = 3) %>%
addResetMapButton()
m
```

**Historical enrollment.** The historical enrollment tab was developed using a custom.csv file that was coded by combining several years of IPEDS data. This dashboard allows the user to select a state and then one or more institutions to track their historic enrollment. Thus, this required a line plot using *ggplotly* and input controls [4,5]. The.csv file was loaded into Shiny. Input controls were developed to allow the user to filter by state and institution. The following code shows how the input controls were programmed in the ui. This is found in the RPubs link in the context of the entire programming for the dashboard. As shown, only one input control for the state filter was coded, but this dashboard includes two input controls (one for state and one for intuition). This is because the second input control was included in the server section and is covered in the next coding example. The following code programs a filter button labeled "First Select a State (Use 'Delete' Key to Dissect)", that it draws from the "State" variable of the "EnrollmentDB" file (notice the "$" symbol). In addition, "WY" will be the default selection and it connects the ui ouput "instInput4".

```
selectInput("stateInput4", "First Select a State (Use 'Delete' Key to
Disselect):",
choices = sort(unique(EnrollmentDB$State)),
selected = "WY", multiple = TRUE),
uiOutput("instInput4"))
```

This next chunk of code shows how the input control to filter for institution was integrated with the input control to filter by state. The *eventReactive()* function uses "input$stateInput4". This tells R to use the "stateInput4" as a link separated by a "$" to filter the data; "stateInput4" was programmed in the ui (and is shown in the previous code example). Each input control was labeled as "df6" and "df7" respectively, and institution selections are limited to the state that is first selected. For example, if one were to select Colorado, only public IPEDS reporting Colorado postsecondary institutions are listed as options in the institution filter.

```
df6 <- eventReactive(input$stateInput4, {%>% filter(State %in% input
$stateInput4)
})
output$instInput4 <- renderUI({selectInput("instInput4", "Next Select
One or More College:", sort(unique(df6()$Institution)),
selected = "Casper College", multiple = TRUE)})
df7 <- eventReactive(input$instInput4, {
df6() %>% filter(Institution %in% input$instInput4)})
```

*Plotly* was used to generate the plot. The code illustrates that the data source used for the plot was "df7" or the input control label. This links the plot to the filters since "df6" is linked within "df7". Aesthetic elements are also included in the code. These plots use the general pattern of code as required for *ggplot2*. By wrapping the plot with *ggplotly*, it provides the user tooltips on the interface when they hover over graph elements [4,5].

```
enroll <-
ggplot(df7(), aes(x = factor(Year), y = Enrollment,
group = Institution, color = Institution, text = paste("Institution:",
Institution, "<br />State:",
State, "<br />Year:", Year, "<br />Enrollment Total:", Enrollment)))+
geom_line(stat = "summary", fun = "mean")+
geom_point(stat = "summary", fun = "mean")+
ggtitle("Fall Enrollment")+
xlab("")+
ylab("Enrollment")+
geom_text(aes(label = Enrollment), position = position_nudge(y = 10),
size = 3, stat = "summary")+
scale_color_brewer(palette = "Dark2")+
theme(axis.text.x = element_text(angle = 45))
ggplotly(enroll, tooltip = 'text')
```

**Demographics.** This tab includes a side bar chart that includes demographic information for each selected institution. The input controls are similar to those in the historical enrollment tab; they are simply given different names. Thus, it is important to know which names match which plots as they use the same dataset. This plot also uses a.csv file that is loaded in Shiny. The following provides the programming code. The input control code is similar to the historic enrollment input control code. For this plot, the data name used is "df9".

```
demo <- ggplot(df9(), aes(x = Demographic, y = Percent,
group = Institution, fill = Institution,
text = paste("Institution:", Institution, "<br />State:", State, "<br
/>Demographic:", Demographic,
"<br />Percent:", Percent %>% paste0("%"))))+
geom_bar(stat = "summary", fun = "mean")+
ggtitle("Demographics")+
xlab("")+
ylab("Percent")+
facet_grid(vars(Institution))+
geom_text(aes(label = paste0(Percent, "%")),
position = position_nudge(y = 3.5), size = 3, stat = "summary")+
scale_y_continuous(labels = function(x) paste0(x, "%"))+
scale_fill_brewer(palette = "Dark2")+
coord_flip()+
theme(axis.text.x = element_text(angle = 45))
ggplotly(demo, tooltip = 'text')
```

**Graduation and retention.** This tab is essentially the same as the demographic tab. It uses a similar interface, similar input controls and programming code, and pulls from a.csv file. It provides graduation and retention information for selected institutions. The code is provided here.

```
grad <- ggplot(df1(), aes(x = RateLevel, y = Rate,
group = Institution, fill = Institution,
text = paste("Institution:", Institution, "<br />State:", State, "<br
/>Rate Level:", RateLevel, "<br />Graduation
Rate:", Rate %>% paste0("%"))))+
geom_bar(stat = "summary", fun = "mean")+
ggtitle("Graduation Rates")+
xlab("")+
```

```
ylab("Rate")+
facet_grid(vars(Institution))+
geom_text(aes(label = paste0(Rate, "%")), position = position_nudge
(y = 1), size = 3, stat = "summary")+
scale_y_continuous(labels = function(x) paste0(x, "%"))+
scale_fill_brewer(palette = "Dark2")+
coord_flip()+
theme(axis.text.x = element_text(angle = 45))
ggplotly(grad, tooltip = 'text')
```

**Dynamic scatter plot.** The dynamic scatterplot provides the correlation between institutional and county factors. Thus, it uses a custom dataset built from the master IPEDS data file and creates a file called "corrdata" and writes it as a.csv file as shown. The code starts by loading a package called *shinyWidgets* [4]. This first chunk of code names a dataset "Community" by selecting the listed variables from the "ipedscensusdata" file. It then encodes it with the *iconv()* function to get rid of accents or characters that will prevent filtering, and then writes it to a.csv file to generate the plot. When programming with public data sets, it is helpful to encode the data, especially when drawing from variable names that might not match R's language.

```
library(shinyWidgets)
Community <-
ipedscensusdata %>%
select(Institution, State, Community_Type, Type, County, FT_Reten-
tion, PT_Retention, TwoYGradRate100, TwoYGradRate150, TwoYGra-
dRate200, FourYGradRate100, FourYGradRate150, FourYGradRate200,
Cost_Off_Campus, Cost_on_Campus, Percent_Women, Percent_FT, Percent_-
White, Median_Household_Income, County_Percent_Veteran, County_Per-
cent_in_Same_House, County_Percent_Never_Married,
County_Percent_Married, County_Percent_Divorced, County_Percent_Sepa-
rated, County_Percent_Widowed, County_Percent_Single, County_Percen-
t_Less_than_HS, County_Percent_HS, County_Percent_Some_or_AS,
County_Percent_Bach, County_Percent_Grad_or_Pro, County_Percent_Sin-
gle_Parent, County_Percent_Not_Citizen, County_Percent_Imigrant,
County_Percent_Rent, County_Percent_Unemployed, County_Percent_White)
Community$County <- iconv(Community$County, from = 'UTF-8', to =
'ASCII//TRANSLIT')
write.csv(Community, "../Dashboard/ShinyFiles/Community.csv", row.
names = FALSE)
```

Drawing from the "Community" file, we write a "corrdata" file to be used in the Shiny application. This may be an unnecessary step as both processes could be combined. We prefer to do some processes iteratively in case steps need to be retraced.

```
corrdata <-
Community %>%
select(Institution, State, Community_Type, Type, County, FT_Reten-
tion, PT_Retention, TwoYGradRate100, TwoYGradRate150, TwoYGra-
dRate200, FourYGradRate100, FourYGradRate150, FourYGradRate200,
Cost_Off_Campus, Cost_on_Campus, Percent_Women, Percent_FT, Percent_-
White, Median_Household_Income, County_Percent_Veteran, County_Per-
cent_in_Same_House, County_Percent_Never_Married,
County_Percent_Married, County_Percent_Divorced, County_Percent_Sepa-
rated, County_Percent_Widowed, County_Percent_Single, County_Percen-
t_Less_than_HS, County_Percent_HS, County_Percent_Some_or_AS,
County_Percent_Bach, County_Percent_Grad_or_Pro, County_Percent_Sin-
gle_Parent, County_Percent_Not_Citizen, County_Percent_Imigrant,
County_Percent_Rent, County_Percent_Unemployed, County_Percent_White)
write.csv(corrdata, "../Dashboard/ShinyFiles/corrdata.csv")
```

These data are then used to generate a scatterplot where the user can select the variables for the X and Y axes, and where each institution is colored as a dot by locale. An additional filter is added to select specific states. The variable selection code is given in the *ggplot2* code for the scatterplot. The code below show how to develop input controls to allow the user to select variables for the *x* and *y* axes of the scatterplot, and the code shows how to program the input control to select state. A picker input control is used for state, which allows the user to check one or more options. This is coded, in the ui portion of the application file.

```
varSelectInput(
inputId = "xvar",
label = "Select an X variable",
data = Community,
selected = "County_Percent_Unemployed"),
varSelectInput(
inputId = "yvar",
label = "Select a Y variable",
data = Community,
selected = "Median_Household_Income"),
pickerInput("stateInput6", "Select a State:",
choices = sort(unique(Community$State)),
options = list('actions-box' = TRUE), multiple = TRUE,
selected = Community$State))
```

The first thing to do is to program and label the state input control that will be linked in the plot. The following provides that code, which is named "ab".

```
ab <- reactive({
Community %>%
filter(State %in% input$stateInput6)
```

An additional feature is added below the scatterplot that gives the regression results of each variable combination given every intuition in the dataset (so it does not change when state if filtered). This feature includes slope, intercept, *p* value and *r-square* statistics. The code is given here. As shown, a basic regression formula is used and the "ab" input control label is used as the data.

```
model <- eventReactive(c(input$xvar, input$yvar), {
req(c(input$xvar, input$yvar))
lm(as.formula(paste(input$yvar, collapse = "+", " ~ ", paste(input
$xvar, collapse = "+"))), data = ab())
```

Finally, *ggplot2* is used to generate the scatterplot. The code is given below. The result is a filterable scatterplot (note where the *x* and *y* input controls are located) with regression results as printed text below the plot [4]. Here *ggplot2* to colors the dots by college type (2-year or 4-year) for demonstration purposes. *Shiny* application developers can choose this feature as it allows the user to filtering by clicking on the legend.

```
com <-
ggplot(ab(), aes_string(x = input$xvar, y = input$yvar))+
geom_point(aes(color = Community_Type, label3 = State,
label4 = County, label5 = Institution))+
geom_smooth(method = "lm")+
scale_color_discrete(name = " ")+
theme(axis.text.x = element_text(angle = 45))
ggplotly(com)
```

**Correlation coefficients.** The next tab of the dashboard allows the user to examine a matrix of correlation coefficients between all the variables. The follow code shows the variables that were loaded into the data set and it also shows how a Pearson's *r* correlation matrix was generated between those variables and written as a data table named "corrmatrix2" and then written into a.csv file.

```
corrdata <- read.csv("../Dashboard/ShinyFiles/corrdata.csv")
corrdata <- select(corrdata, Institution, State, Community_Type,
Type, County, FT_Retention, PT_Retention, TwoYGradRate100, TwoYGra-
dRate150, TwoYGradRate200, FourYGradRate100, FourYGradRate150, FourY-
GradRate200, Cost_Off_Campus, Cost_on_Campus, Percent_Women,
Percent_FT, Percent_White, Median_Household_Income, County_Percent_-
Veteran, County_Percent_in_Same_House, County_Percent_Never_Married,
County_Percent_Married, County_Percent_Divorced, County_Percent_Sepa-
rated, County_Percent_Widowed, County_Percent_Single, County_Percen-
t_Less_than_HS, County_Percent_HS, County_Percent_Some_or_AS,
County_Percent_Bach, County_Percent_Grad_or_Pro, County_Percent_Sin-
gle_Parent, County_Percent_Not_Citizen, County_Percent_Imigrant,
County_Percent_Rent, County_Percent_Unemployed, County_Percent_White)
corrmatrix <-
round(cor(corrdata[sapply(corrdata, is.numeric)], use = 'pairwise'),
2)
write.csv(corrmatrix, "../Dashboard/ShinyFiles/corrmatrix2.csv")
```

This next section of code provides a method of shading the cells in the correlation matrix depending on the magnitude of the correlation using the *corRampPalette*() function [39].

```
brks <- seq(-1, 1, .01)
clrs <- colorRampPalette(c("white", "#6baed6"))(length(brks) + 1)
dataCol_df <- ncol(corrmatrix) - 1
dataColRng <- 1:dataCol_df
```

After setting the color preferences given correlation coefficient, a table was programmed where the user was able to select the X and Y variables and custom build the table where the coefficients would be shaded according to the strength of the correlation using the DT library [36]. The linked RPubs page gives the complete programming sequence.

```
server <- function(input, output, session){
varfilter <- reactive({
filtered <- corrmatrix %>%
filter(variable %in% input$varInput)
})
output$corrtable <- DT::renderDataTable(datatable({
if (length(input$columnInput) = = 0) return(varfilter())
varfilter() %>%
dplyr::select(!!!input$columnInput)
}, rownames = TRUE, extensions = "FixedColumns",
options = list(paging = TRUE, searching = FALSE, info = FALSE,
sort = TRUE, scrollX = TRUE, fixedColumns = list(leftColumns = 2))) %>
%
formatStyle(columns = dataColRng, backgroundColor = styleInterval
(brks, clrs)))
}
```

**Degrees and careers.** The degrees and careers tab includes a table with picker input controls that allow the user to select multiple states, institutions, and degrees. The input controls are linked so they limit each other's selection choices when one or more input control is selected. The data for this table is derived from the CIP/SOC data table that was generated in RMarkdown and exported to the folder with the Shiny application file. The following code shows how the picker input controls were generated in the ui.

```
pickerInput("stateInputd", "Select or type one or more states",
choices = sort(unique(Degrees_and_Jobs$Select_a_State)),
options = list('actions-box' = TRUE),
multiple = TRUE,
selected = "AK"),
pickerInput("instInputd", "Select or type one or more institutions:",
```

```
choices = sort(unique(Degrees_and_Jobs$Institution_Name)),
options = list('actions-box' = TRUE),
multiple = TRUE),
pickerInput("degreeInput", "Select or type one or more degrees:",
choices = sort(unique(Degrees_and_Jobs$Degree_Title)),
options = list('actions-box' = TRUE),
multiple = TRUE)
```

The following program language shows how the input controls link in the server section of the Shiny application file. The logic of code is assigned the name of "state-deg", which is used to render the data table, linking the picker input controls and their desired behaviors to the table. The table is rendered as output using the *DT:: RenderDataTable()* function [36]. The result is a dashboard with a table of degree titles linked to careers and income data that is filterable by state, institution, and degree.

```
state_deg <- reactive({
filter(Degrees_and_Jobs, Select_a_State %in%input$stateInputd)
})
observeEvent(state_deg(), {
choices <- sort(unique(state_deg()$Institution_Name))
updatePickerInput(session = session, inputId = "instInputd",
choices = choices, selected = Degrees_and_Jobs$Institution_Name)
})
institution_deg <- reactive({
req(input$instInputd)
filter(state_deg(), Institution_Name %in% input$instInputd)
})
observeEvent(institution_deg(), {
choices <- sort(unique(institution_deg()$Degree_Title))
updatePickerInput(session = session, inputId = "degreeInput",
choices = choices, selected = Degrees_and_Jobs$Degree_Title)
})
output$degrees <- DT::renderDataTable(options = list
(autoWidth = TRUE, scrollX = TRUE, searching = FALSE), {
req(input$degreeInput)
institution_deg() %>%
filter(Degree_Title %in% input$degreeInput) %>%
select(Institution_Name, Degree_Title, Career_Title, Mean_Hourly,
Mean_Annual, Hourly_Median, Median_Annual)
})
}
```

**Degrees and job projections.** This table and its linked input controls was programmed using similar code as the degrees and careers table on the previous tab menu [36]. This table also includes linked picker input controls to limit the data in the table to states, institutions, a degree titles. This table provides the user with information about the careers linked to each institution's degrees. The information provided includes number of jobs, projected number of jobs in ten years (between 2019 and 2029), percent growth (or decline) in job availability between 2019 and 2029, annual average job openings, wage information, and education and job training required to enter the field.

**Data dictionary.** Microsoft Excel was used to manually enter all the variables used in the dashboard including the variable's name as it is used, the data source of that variable, and a description of the variable. This dictionary was developed to help the users better understand the elements of the dashboard. The Microsoft Excel file was saved as a.csv in the folder with the Shiny application file, loaded into the application and included as the last tab of the dashboard with the following simple code [36].

```
output$dictionarytable <- DT::renderDataTable(datadictionary,
options = list(scrollX = TRUE))
```

**Deployment of the dashboard.** This dashboard was generated on free and open-source software provided by RStudio (now known as Posit) using RMarkdown to munge the data and Shiny to deploy it. To deploy the dashboard, we first recommend running the Shiny application by hitting the "Run" button. Shiny will process the code and give specific errors that can be looked up and located by line number. Once you trouble shoot and the application runs, go through it tab by tab and make sure it's functional. After it is satisfactory, press "publish". If you don't have a Shiny account yet, it will ask you to set one up. Shiny offers good directions on how to link your account to your application and also how to deploy it here: https://shiny.rstudio.com/deploy/ [40].

Shiny offers a free account with a limit of five applications and 25 active hours per month. If a Shiny application becomes more popular with more user hits, there are options to upgrade to more applications (i.e. dashboards) and more hours [41]. R also offers other tools such as flexdashboard to develop interactive dashboards [42]. These other tools may require similar data munging code, but may require different coding structure for dashboard design. The purpose of this paper was to introduce the reader to Shiny because of its free resources and versatility.

**Updates and maintenance.** Maintaining the dashboard and keeping the data current requires annual activities. First, one must understand the data collection and refresh schedule of each of the data sources. IPEDS updates its data early every summer [8]. The United States Census Bureau (USCB) updates around the beginning of the fiscal year, or July 1 [23], and the Bureau of Labor Statistics (BLS) updates their wage data quarterly on the 12[th] of March, June, September, and December [19]. To maintain this particular dashboard, we recommend updating mid-July, though one could update BLS data more frequently without updating IPEDS or US Census data.

To update USCB data, simply change the programming language to the desired date. For example, instead of 2020, change it to 2021. This is achieved by meticulously going through the code and changing each dated variable or table, sometimes with a search and replace. To update IPEDS, first obtain the latest Access file and then update the years of your tables and variables. For example, the "HD2019" table will change to "HD2020". BLS data can be updated by obtaining the latest data from their website. If you name the new data file the same as the previous one, it should load and run, but be sure to first check the variable names and other details of the file. Once you feel your data are refreshed, run the application and go through your updated dashboard and randomly check numbers and functions for errors before deploying it. It is also a good idea to have somebody else look at it with fresh eyes, preferably somebody with content expertise of your dashboard.

## Discussion

The information available in federal datasets is invaluable in providing stakeholders and analysts information. Despite the availability of national data through systems such as the USCB, BLS, and IPEDS, people may find it difficult to navigate the vast choices and spreadsheets to make these data useful for research and decision making. User friendly data dashboards that join multiple public data sets can be used by researchers, data scientists, analysts, administrators, and other professionals to inform decision making and policy and engage in continuous improvement in their respective fields Though some research has combined the USCB, BLS, and IPEDS datasets [22,25,26,31], no examples of user interface dashboards are currently present in the literature. This paper provided an overview of a dashboard developed using Shiny

[4] that combines data from NCES's IPEDS system with the USCB and the BLS allowing the user to explore many potential research questions from the basic descriptive level to more complex and inferential. This paper provided a summary of the data collection, coding methodology, and general layout of this dashboard and its development using Shiny in hopes that other programmers and parties interested in combining and using these types of data can learn how to develop and use them within a navigable dashboard. The demonstrated dashboard was created to allow higher education administrators, institutional researchers, and postsecondary policymakers to select institutions, learn their costs and enrollments, check their historic enrollment and compare them to others, investigate demographic and outcomes data and make comparisons between institutions, examine the correlations with institutional factors using IPEDS variables and community factors using USCB variables, and examine BLS career income and projections by institutional degrees. Other programmers and researchers have the potential to develop a variety of dashboards using similar public data to help inform decision making and policy.

## Supporting information

**S1 Table. R package libraries used to develop the project.**
(DOCX)

## Author Contributions

**Conceptualization:** Mark A. Perkins.

**Data curation:** Mark A. Perkins.

**Formal analysis:** Mark A. Perkins.

**Methodology:** Mark A. Perkins.

**Project administration:** Mark A. Perkins.

**Resources:** Mark A. Perkins.

**Software:** Mark A. Perkins.

**Writing – original draft:** Mark A. Perkins, Jonathan W. Carrier.

**Writing – review & editing:** Mark A. Perkins, Jonathan W. Carrier.

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
