## [Decision Letter · Decision Letter 0]

11 Mar 2022

PONE-D-21-37932Using RShiny to Develop a Dashboard using IPEDS, U.S. Census, and Bureau of Labor Statistics DataPLOS ONE

Dear Dr. Perkins,

Thank you for submitting your manuscript to PLOS ONE. After careful consideration, we feel that it has merit but does not fully meet PLOS ONE’s publication criteria as it currently stands. Therefore, we invite you to submit a revised version of the manuscript that addresses the points raised during the review process.

Please submit your revised manuscript within two months. If you will need more time than this to complete your revisions, please reply to this message or contact the journal office at plosone@plos.org. Please include the following items when submitting your revised manuscript:A rebuttal letter that responds to each point raised by the academic editor and reviewer(s). You should upload this letter as a separate file labeled 'Response to Reviewers'.A marked-up copy of your manuscript that highlights changes made to the original version. You should upload this as a separate file labeled 'Revised Manuscript with Track Changes'.An unmarked version of your revised paper without tracked changes. You should upload this as a separate file labeled 'Manuscript'.

We look forward to receiving your revised manuscript.

Kind regards,

Barbara Szomolay

Academic Editor

PLOS ONE

Journal Requirements:

2. Please note that PLOS ONE has specific guidelines on software sharing (http://journals.plos.org/plosone/s/materials-and-software-sharing#loc-sharing-software) for manuscripts whose main purpose is the description of a new software or software package. In this case, new software must conform to the Open Source Definition (https://opensource.org/docs/osd) and be deposited in an open software archive. Please see http://journals.plos.org/plosone/s/materials-and-software-sharing#loc-depositing-software for more information on depositing your software.

4. We note that you have stated that you will provide repository information for your data at acceptance. Should your manuscript be accepted for publication, we will hold it until you provide the relevant accession numbers or DOIs necessary to access your data. If you wish to make changes to your Data Availability statement, please describe these changes in your cover letter and we will update your Data Availability statement to reflect the information you provide

Reviewers' comments:

Reviewer's Responses to Questions

**Comments to the Author**

1. Is the manuscript technically sound, and do the data support the conclusions?

Reviewer #1: No

Reviewer #2: Yes

2. Has the statistical analysis been performed appropriately and rigorously? 

Reviewer #1: Yes

Reviewer #2: N/A

3. Have the authors made all data underlying the findings in their manuscript fully available?

Reviewer #1: Yes

Reviewer #2: Yes

4. Is the manuscript presented in an intelligible fashion and written in standard English?

Reviewer #1: Yes

Reviewer #2: Yes

5. Review Comments to the Author

Reviewer #1: I thank the authors for the opportunity to read their paper. Overall, I appreciate their contribution and all the effort placed in the development of this dashboard. I agree with their assessment, so far, I am not aware of any interface dashboards using IPEDS, USCB, and BLS. In the following lines I offer some reactions but overall, I agree with their coding schemes.

My main feedback would be to clarify the purpose and scope of the study. Their dashboard is functional, but it is not clear why would this paper need to be published at PONE? Are the authors interested in teaching researchers to develop similar applications? Are they interested in easing access to these data sources? If the latter is true, I recommend adding a functionality to download the data merged/compiled from their platform. This would be useful for the incorporation of these indicators from a multiplicity of data sources. I can definitely see master’s students taking advantage of this resource.

Other questions that emerge are, why are their merging approaches limited to two institutional types (public 2- and 4-year colleges)?

I am also wondering if the authors have considered relying on the college scorecard (https://collegescorecard.ed.gov/data) data for this resource has been standardized and compiles other data sources. This comment is based on the fact that the IPEDS data need to be standardized when conducting cross-sectional analyses. Although I understand that the current application relies on one academic year, feeding the application from the college scorecard may be an nice addition.

In sum, I believe that the ability to download these data may be an important addition to their dashboard.

Reviewer #2: I found this application very interesting and I applaud the effort to publish the methods behind developing this {shiny} application. These types of dashboards are vital for the research community and often do not receive the academic credit they deserve. Below are some of my comments and concerns.

1. I'm not sure the purpose of the paper is entirely clear: is it to describe how someone might create a dashboard like this? Or is it to describe the contents of this particular dashboard? At the moment it seems like a bit of a blend, but I found that hard to follow since the audience for the two uses would likely be quite different. If possible, I would try to narrow the focus to one or the other. If trying to describe *how* someone could build a dashboard like this, there needs to be more focus on how to develop a shiny dashboard. For example, the basic concepts like the `ui` and `server` were not well explained in the text. Similarly the `input$` `output$` system in {shiny} was not explained. Additionally, if this is the focus, less is needed on the background about the particular datasets chosen to integrate here (since presumably the next user would not be building this exact dashboard), and more attention could be paid to the general concepts of downloading, munging, and merging data together (with these datasets as a particular example, but not the main focus). If, on the other hand, the purpose of this paper is to *describe* the contents of this particular dashboard, then the focus on the code is not as necessary.

2. It is not entirely clear to me what the purpose of this application is. Is it just to allow users to explore this data? If it is for analysis (but presumably for people who are less keen on pulling all of the data themselves) it seems like it would need to have the ability to subset the large datasets for download. Currently, it seems all the user can do is calculate the correlation between two variables or compare a few variables between institutions.

3. The R code was hard to read and a bit inconsistent style-wise. I recommend using a linter to keep the code consistent (for example: https://github.com/r-lib/lintr)

4. I'm not sure "RShiny" is how RStudio would refer to this product (I think just Shiny for the product {shiny} for the package?).

5. Since the purpose of the paper is not entirely clear, I'm not sure if this review includes a review of the dashboard itself, but if so, here are a few comments:

* On the "Institution map" page it says to "use the search box", however I do not see a search box on this page (other than the magnifying glass on the map itself?)

* On the "Historic Enrollment" page, the x-axis could be cleaned up to just say 2015, 2016, 2017, 2018 (instead of `2015_Enrollment` etc.). Additionally, when more than one school is added, the numbers are completely obsured by the points on the graph. Either the points could be removed (and replaced with just the numbers), the numbers could be moved up a bit, or the user could just hover to see the numbers.

* On the "Demographics" page, the y-axis could be cleaned up (to remove the underscore and also the word "Percent" since it is redundant with the x-axis). The ordering of the bars could be improved (maybe ordered by frequency based on the top selection?) to make it easier to read for a viewer. It is also hard to compare categories between institutions - maybe position_dodge rather than a facetted chart would make sense? Otherwise, perhaps flipping the direction so the categories are aligned across institutions to make it easier to compare between them. (The same for the Graduation and Retention tab)

* In the "Dynamic Scatterplot" page, it is not clear why the points are colored by location type? It also would be nice to have a direct link on this page to the Data Dictionary since that is necessary to know what the X and Y variables are. Additionally, it would be nice to output the results in a Tabel rather than the `lm` output directly, if possible, since the intended audience is likely not familiar with R (it seems).

* In the "Correlation Coefficients" page "variable" is forced to stay in the selection. It is trivial to force this in the dataset without relying on the user to not delete it (create a string of the names to pass to the selectizeInput that removed "variable" from the choice and similarly when picking the variables on the server side add "variable" back in). It would also be nice to have a direct link to the Data Dictionary from this page.

6. PLOS authors have the option to publish the peer review history of their article (what does this mean?). If published, this will include your full peer review and any attached files.

Reviewer #1: No

Reviewer #2: No

---

## [Author Response · Author response to Decision Letter 0]

29 Apr 2022

April 29, 2022

PLoS One Editorial Board

1265 Battery Street

Suite 200

San Francisco, CA 94111

To the Editors and Peer-Reviewers:

This letter is regarding manuscript PONE-D-21-37932R1. First, we would like to thank the editors and reviewers for their time and focus on our paper titled, “Using RStudio to Develop a Dashboard using IPEDS, U.S. Census, and Bureau of Labor Statistics Data.” We would like to emphasize that this letter does not dispute any of the feedback we received on the paper or on the dynamic dashboard application. In fact, this letter acknowledges all suggestions and edits and addresses how each were approached in the attached revised manuscript. Further, we would like to note that the feedback was greatly appreciated as we are always looking for ways to improve this work and have found submitting to PLoS One rewarding because of the feedback we have received. Therefore, we give our sincerest thanks to the reviewers and editors for the time and thought they put into examining our work.

Second, we would like to state that we have created an RPubs page on which to share our programming and code. The submitted version of the manuscript and the updated RStudio application both contain links to this page where the reader will be able to view all of the programming code that was used to generate the dashboard. We mention this because this will not require any accession numbers or DOIs. All available data and programming language will be accessible through the application, or through the link to RPubs which is here: https://rpubs.com/IPEDS

In addition, we have included download buttons on the application itself where the user can download all datasets from the application. The combination of the programming code and the data download buttons allows for open-source data access.

We found all of your feedback valuable and used it to improve our paper and the dashboard itself. The following provides point-by-point overview of our response to your feedback. 

Formatting feedback from editor:

Please ensure that your manuscript meets PLOS ONE's style requirements, including those for file naming. The PLOS ONE style templates can be found at https://journals.plos.org/plosone/s/file?id=wjVg/PLOSOne_formatting_sample_main_body.pdf and https://journals.plos.org/plosone/s/file?id=ba62/PLOSOne_formatting_sample_title_authors_affiliations.pdf

Response:

We reformatted the manuscript to use Vancouver Brackets style of formatting and followed the PLOS guidelines.

Software feedback from editor:

Please note that PLOS ONE has specific guidelines on software sharing (http://journals.plos.org/plosone/s/materials-and-software-sharing#loc-sharing-software) for manuscripts whose main purpose is the description of a new software or software package. In this case, new software must conform to the Open Source Definition (https://opensource.org/docs/osd) and be deposited in an open software archive. Please see http://journals.plos.org/plosone/s/materials-and-software-sharing#loc-depositing-software for more information on depositing your software.

Response:

This paper presents a web application that was programmed using R open-source software. We reviewed this policy and find it to be open source and available through a publicly accessible web address.

Software feedback from editor:

We note that you have stated that you will provide repository information for your data at acceptance. Should your manuscript be accepted for publication, we will hold it until you provide the relevant accession numbers or DOIs necessary to access your data. If you wish to make changes to your Data Availability statement, please describe these changes in your cover letter and we will update your Data Availability statement to reflect the information you provide.

Response:

We have included a link to the data in the application and a link to the programming code to get the data on an RPubs web page. Both the application and the web page are publicly accessible. We also address this in a draft of our new cover letter and in the manuscript.

Reviewer 1 Comment 1:

My main feedback would be to clarify the purpose and scope of the study. Their dashboard is functional, but it is not clear why would this paper need to be published at PONE? Are the authors interested in teaching researchers to develop similar applications? Are they interested in easing access to these data sources? If the latter is true, I recommend adding a functionality to download the data merged/compiled from their platform. This would be useful for the incorporation of these indicators from a multiplicity of data sources. I can definitely see master’s students taking advantage of this resource.

Response:

We identified with this feedback as we grappled with that question ourselves, and after much discussion, have decided to emphasize the main purpose of the paper as a demonstration of how to develop the application including data querying, munging, and programming. You will see this change of focus in the revised manuscript. We also emphasize that the programming and approach may be applicable to other datasets.

Reviewer 1 Comment 2:

Why are their merging approaches limited to two institutional types (public 2- and 4-year colleges)?

Response:

We explained how and why two categories were used and that other programmers could code more by studying the accompanying website with our code on it. We coded two categories because IPEDs generally classifies institutions with several levels and by reducing the code to two categories we are able to demonstrate recoding programming language. Again, the reader could use this to code any number of categories desired. We point this out in the paper as well.

Reviewer 1 Comment 3:

I am also wondering if the authors have considered relying on the college scorecard (https://collegescorecard.ed.gov/data) data for this resource has been standardized and compiles other data sources. This comment is based on the fact that the IPEDS data need to be standardized when conducting cross-sectional analyses. Although I understand that the current application relies on one academic year, feeding the application from the college scorecard may be a nice addition.

Response:

This is an excellent resource and many of the data elements available (e.g. ACT/SAT scores) on the scorecard are actually (and coincidentally) the source for a project in the works to develop an additional dashboard to examine those data. We considered these data when first developing this project and agree that the college scorecard is an excellent resource that does use some of the IPEDs data elements that we used in our analysis. The most recent data dictionary of the scorecard notes that many of the Treasury elements have been discontinued. We found it easier to use the tidycensus package to get county level elements regarding income and other demographics. In addition, we note your comment about standardization and did not find that necessary for this project as we just needed descriptive values for our outputs. However, our future project (as noted above) will need this and thus requires us to consider these data elements. We are specifically interested in the rscorecard package and API and that project is under way. This will lead to a later paper and project.

Reviewer 2 Comment 1:

I'm not sure the purpose of the paper is entirely clear: is it to describe how someone might create a dashboard like this? Or is it to describe the contents of this particular dashboard? At the moment it seems like a bit of a blend, but I found that hard to follow since the audience for the two uses would likely be quite different. If possible, I would try to narrow the focus to one or the other. If trying to describe *how* someone could build a dashboard like this, there needs to be more focus on how to develop a shiny dashboard. For example, the basic concepts like the `ui` and `server` were not well explained in the text. Similarly the `input$` `output$` system in {shiny} was not explained. Additionally, if this is the focus, less is needed on the background about the particular datasets chosen to integrate here (since presumably the next user would not be building this exact dashboard), and more attention could be paid to the general concepts of downloading, munging, and merging data together (with these datasets as a particular example, but not the main focus). If, on the other hand, the purpose of this paper is to *describe* the contents of this particular dashboard, then the focus on the code is not as necessary.

Response:

The first reviewer’s concern aligns with that of the second reviewer. Therefore, re-clarified the purpose of this dashboard in the revised manuscript. We do provide an overview of the datasets for context, but we emphasize that the user could apply this to other datasets. 

We also went through the paper and made significant additions in the methods section to describe many of the functions and the logic of shiny. We also added the entirety of our programming code as a link.

Reviewer 2 Comment 2:

It is not entirely clear to me what the purpose of this application is. Is it just to allow users to explore this data? If it is for analysis (but presumably for people who are less keen on pulling all of the data themselves) it seems like it would need to have the ability to subset the large datasets for download. Currently, it seems all the user can do is calculate the correlation between two variables or compare a few variables between institutions.

Response:

We have decided to focus the emphasis on how to program dashboards using publicly available data sets. We retain information about the datasets used but articulate that our methods are applicable to other datasets. We also included the ability to pull the data and examine the code if the user desires to use it for their own purposes. We also demonstrate how to include a download handler in the programming of the application.

Reviewer 2 Comment 3:

The R code was hard to read and a bit inconsistent style-wise. I recommend using a linter to keep the code consistent (for example: https://github.com/r-lib/lintr)

Response:

We ran both the data munging file and the application through lintr and addressed several formatting details given the specific suggestions of Hadley Wickham. This helped us clean up our code on such details as visible binding for global variables, spacing issues (particularly around infix operators), trailing blank spaces, and trailing white spaces. We incorporated these changes to the paper, but more importantly, wrote them on the RPubs page using knitr on RMarkdown. We did not address 80 character spaces in our HTML code, and we did not make all variable and function names snake case, the latter mainly for functionality purposes. In addition, we utilized page-break to keep the chunks of code together so they don’t split pages. We did not use track changes when adjusting the code in the revised manuscript as we felt it was too confusing.

Reviewer 2 Comment 4:

I'm not sure "RShiny" is how RStudio would refer to this product (I think just Shiny for the product {shiny} for the package?).

Response:

We have adjusted this throughout the manuscript.

 

Reviewer 2 Comment 6:

Since the purpose of the paper is not entirely clear, I'm not sure if this review includes a review of the dashboard itself, but if so, here are a few comments:

Response:

We addressed all of these comments in the following list. These were very helpful in improving our application.

• On the "Institution map" page it says to "use the search box", however I do not see a search box on this page (other than the magnifying glass on the map itself?)

o This is a great observation and our original directions could well have been confusing to users. We have therefore changed the text to say “magnifying glass” instead of “search box”.

• On the "Historic Enrollment" page, the x-axis could be cleaned up to just say 2015, 2016, 2017, 2018 (instead of `2015_Enrollment` etc.). Additionally, when more than one school is added, the numbers are completely obscured by the points on the graph. Either the points could be removed (and replaced with just the numbers), the numbers could be moved up a bit, or the user could just hover to see the numbers.

o The dates have been addressed by recoding the value labels in the dataset. We agree that this improves the look and feel of the application.

o The obscured numbers was an excellent observation. We eliminated this problem as suggested by eliminating them and leaving it to the tool tip and examination of the axis. 

• On the "Demographics" page, the y-axis could be cleaned up (to remove the underscore and also the word "Percent" since it is redundant with the x-axis). The ordering of the bars could be improved (maybe ordered by frequency based on the top selection?) to make it easier to read for a viewer. It is also hard to compare categories between institutions - maybe position_dodge rather than a facetted chart would make sense? Otherwise, perhaps flipping the direction so the categories are aligned across institutions to make it easier to compare between them. (The same for the Graduation and Retention tab) 

o The y-axis labels have been cleaned as suggested and it is much improved.

o We used “position = ‘dodge2’” and replaced that with the facet_grid. We also got rid of the numbers like the enrollment report and kept them in the tooltip. This feedback significantly improved the graph.

o We tried to order by frequency by using the reorder() function and that worked well with one institution, but we would get an error when the maximum category was not in agreement between institutions. For example, if the highest was percent women for institution A but the highest was white for institution B, then it would generate an error.

• In the "Dynamic Scatterplot" page, it is not clear why the points are colored by location type? It also would be nice to have a direct link on this page to the Data Dictionary since that is necessary to know what the X and Y variables are. Additionally, it would be nice to output the results in a Table rather than the `lm` output directly, if possible, since the intended audience is likely not familiar with R (it seems). 

o We created a download button to get the data dictionary on this page. 

o We were unable to put the output results in a table. However, we re-coded the text and simplified the results so that the r-squared values were much easier to read and the intercepts and coefficients were cleaner without all the programming code in the way to distract. 

o However, with the adjustments with position dodge, this is improved.

• In the "Correlation Coefficients" page "variable" is forced to stay in the selection. It is trivial to force this in the dataset without relying on the user to not delete it (create a string of the names to pass to the selectizeInput that removed "variable" from the choice and similarly when picking the variables on the server side add "variable" back in). It would also be nice to have a direct link to the Data Dictionary from this page. 

o We implemented code to eliminate this from the fixed variable and forced an unfilterable string variable as row labels using this coding logic: rownames(corrmatrix) <- corrmatrix$variable

o This was an element that bothered us, so we are glad that it was pointed out.

o We included a button to download the data dictionary as well as a link.

o We also included this coding logic in the manuscript and on the RPubs site.

Thank you once again for your feedback.

Sincerely,

Authors

---

## [Decision Letter · Decision Letter 1]

4 Oct 2022

PONE-D-21-37932R1Using RStudio to Develop a Dashboard using IPEDS, U.S. Census, and Bureau of Labor Statistics DataPLOS ONE

Dear Dr. Perkins,

Thank you for submitting your manuscript to PLOS ONE. After careful consideration, we feel that it has merit but does not fully meet PLOS ONE’s publication criteria as it currently stands. Therefore, we invite you to submit a revised version of the manuscript that addresses the points raised during the review process.

We look forward to receiving your revised manuscript.

Kind regards,

Sathish A.P. Kumar

Academic Editor

PLOS ONE

Journal Requirements:

Reviewers' comments:

Reviewer's Responses to Questions

**Comments to the Author**

1. If the authors have adequately addressed your comments raised in a previous round of review and you feel that this manuscript is now acceptable for publication, you may indicate that here to bypass the “Comments to the Author” section, enter your conflict of interest statement in the “Confidential to Editor” section, and submit your "Accept" recommendation.

Reviewer #2: (No Response)

Reviewer #3: (No Response)

2. Is the manuscript technically sound, and do the data support the conclusions?

Reviewer #2: Yes

Reviewer #3: Yes

3. Has the statistical analysis been performed appropriately and rigorously? 

Reviewer #2: N/A

Reviewer #3: Yes

4. Have the authors made all data underlying the findings in their manuscript fully available?

Reviewer #2: Yes

Reviewer #3: Yes

5. Is the manuscript presented in an intelligible fashion and written in standard English?

Reviewer #2: Yes

Reviewer #3: Yes

6. Review Comments to the Author

Reviewer #2: # Review

I appreciate the updates, I have a few additional comments, please see below.

1. In the abstract, it looks like "R Shiny" was replaced with "RStudio" -- this may have been due to my previous comment be unclear -- I think this should say "R shiny dashboard", not "RStudio dashboard"

2. There are inconsistencies in how Shiny is referred to (for example RShiny, R shiny, RStudio, shiny etc) I would recommend replacing these all with "Shiny" when referring to the application and "shiny" when referring to the package.

3. The detailed background on the datasets does not seem necessary, particularly the historical context. The information on what the datasets provide, however, seems relevant.

4. I found the following explanation of a pipe confusing:

"You’ll also notice the use of “%>%” in the code. That is a pipe that basically tells R to use the dataset that was previously stated, or to continue processing given what was done before."

Perhaps something like:

"The %>%, known as the "pipe" is a function from the magrittr package. It takes the object on the left hand side and "pipes" it into the first argument of the subsequent function. For example institutioninformation %>% select(UNITID) is equivalent to select(institutioninformation, UNITID). A benefit of the pipe is it can allow the code to be more readable than a series of nested functions."

5. Perhaps explans what the tidyverse is, i.e. "Once the data are queried, it may be necessary to recode variables using tidyverse, a suite of R packages used to manipulate data frames".

6. There are several times the authors refer to assigning an object in R as "names the table" -- technically this should be something like "a table is created called x" since the process is actually creating the table and naming the object something rather than just assigning a name.

7. The code on page 17 could be reduced to a series of left joins connected by the pipe (rather than creating something named "census" over and over again, ie:

census <- left_join(employ, Health, by = "GEOID") %>%

rename(GEOID = GEOID.x) %>%

left_join(income, by = "GEOID") %>%

left_join(TotalWhite, by = "GEOID") %>%

left_join(Veteran, by = "GEOID")

The same is true for all of the code on pages 18 and 19. It seems unnessesary to use the pipe if you aren't going to chain the functions together (I like the pipe, but if you are using it I would recommend coding like above).

8. In the getting started with shiny section, instead of saying you need to open a "shiny" file (this is not a file type), I would say you need to create a .R file.

Small typos:

1. When referring to a function in text, it generally should have the open and closed brackets (or no brackets at all) (i.e obcDriverConnect( should be obcDriverConnect() or obcDriverConnect)

2. Page 15 look says "install. Packages" should be "install.packages"

3. Page 22 "page gives the complete programming language" should read "page gives the complete programming code"

4. Page 25 HTLM -> HTML

Reviewer #3: The authors are to be commended on a producing a polished, well-thought out shiny application using R.

The manuscript provides a discussion of how the authors coded a dashboard and what coding choices facilitated this.

I have a few questions that if answered in the text would strengthen this manuscript as a resource for individuals who would replicate the development of these dashboards:

1. What process did the authors use to select these variables and not others to appear in the dashboards? There are many datapoints left out.

2. A main claim in the paper is that there isn’t a dashboard cited in the literature that combines US federal census, labor and education. Are there any studies that combine this data themselves without a dashboard? This would strengthen the argument that this dashboard was needed and provide a basis to evaluate the effectiveness of this dashboard in future years.

3. Some discussion is needed about the logistics of hosting and maintaining this dashboard. The authors seem to be using a free account on shinyapps.io. What are the limitations as the dashboard is used more frequently for this approach? Did the authors consider other deployment paths? (i.e. was cost the only factor, or were other deployment options evaluated).

4. What plans are there to maintain the data set and what is the workload? i.e. what effort is needed to add 2020 data and beyond? How long will that take given the initial code is in place?

The response to a prior reviewer comment about 2-year or 4-year institutions I think misses the mark. This is a somewhat misleading variable in IPEDS that actually can lead to wrong conclusions if a researcher is unaware what it represents. IPEDs codes institutions as 4-year institutions if they have any bachelor’s degree programs. 2-year institutions (in the IPEDS data set) are those institutions with only associates degrees. Most times this is not actually what a researcher actually wants and excludes nearly all community & technical colleges in some states that award primarily associates degrees and less than two year certificates, but have a few bachelor’s degree programs.

7. PLOS authors have the option to publish the peer review history of their article (what does this mean?). If published, this will include your full peer review and any attached files.

Reviewer #2: No

Reviewer #3: No

---

## [Author Response · Author response to Decision Letter 1]

25 Oct 2022

October 23, 2022

PLOS One Editorial Board

1265 Battery Street

Suite 200

San Francisco, CA 94111

To the Editors and Peer-Reviewers:

We would like to thank you all for the time and attention you have given to our project and paper. We have reviewed your recommendations and feedback carefully and feel we now have a much better paper and dashboard. You will find our responses to your feedback in this document.

Responses to Feedback

Journal Requirements:

Response:

We have reviewed all our references and found nothing retracted. We only had to change the order. This is tracked. 

Reviewer 2 

1) In the abstract, it looks like "R Shiny" was replaced with "RStudio" -- this may have been due to my previous comment be unclear -- I think this should say "R shiny dashboard", not "RStudio dashboard"

Response:

We changed it to say “Using R Shiny to Develop a Dashboard . . .”

2) There are inconsistencies in how Shiny is referred to (for example RShiny, R shiny, RStudio, shiny etc) I would recommend replacing these all with "Shiny" when referring to the application and "shiny" when referring to the package.

Response:

We went through it and changed all of this to Shiny.

3) The detailed background on the datasets does not seem necessary, particularly the historical context. The information on what the datasets provide, however, seems relevant.

Response:

We removed historical content from the document. 

 

4) I found the following explanation of a pipe confusing:

"You’ll also notice the use of “%>%” in the code. That is a pipe that basically tells R to use the dataset that was previously stated, or to continue processing given what was done before."

Perhaps something like:

"The %>%, known as the "pipe" is a function from the magrittr package. It takes the object on the left hand side and "pipes" it into the first argument of the subsequent function. For example institutioninformation %>% select(UNITID) is equivalent to select(institutioninformation, UNITID). A benefit of the pipe is it can allow the code to be more readable than a series of nested functions."5)

Response:

We really liked how you put that and adapted your words. This was very helpful. You’ll notice we wrote:

“The “%>% is known as a pipe and is a function of the magrittr package. This function takes the object on the left hand side and “pipes” it onto the first argument of the subsequent function. For example, institutioninformation %>%select(UNITID) is equivalent to select(institutioninformation, UnitID). The pipe allows the code to be more readable than a series of nested functions.”

5) Perhaps explain what the tidyverse is, i.e. "Once the data are queried, it may be necessary to recode variables using tidyverse, a suite of R packages used to manipulate data frames".

Response:

We adopted your language here again, changing only one word:

“Once the data are queried, it may be necessary to recode variables using tidyverse, a suite of R packages used to clean and munge data frames”

6) There are several times the authors refer to assigning an object in R as "names the table" -- technically this should be something like "a table is created called x" since the process is actually creating the table and naming the object something rather than just assigning a name.

Response:

We went through the paper and changed the language accordingly. 

7) The code on page 17 could be reduced to a series of left joins connected by the pipe (rather than creating something named "census" over and over again, ie:

census <- left_join(employ, Health, by = "GEOID") %>%

rename(GEOID = GEOID.x) %>%

left_join(income, by = "GEOID") %>%

left_join(TotalWhite, by = "GEOID") %>%

left_join(Veteran, by = "GEOID")

Response:

The code has been changed to this. As a side note, we started piping our newest code this way. We used to do it the longer way to detect errors better, but decided your suggestion was better. 

We also fixed this with the IPEDS data joins. 

ipedsdashdata <- left_join(institutioninformation, enrollmentinformationgender, by = "UNITID")%>%

left_join(enrollmentinformationrace, by = "UNITID")%>%

etc. 

We also changed it in the RMarkdown and republished it.

 

8) Small typos:

a. When referring to a function in text, it generally should have the open and closed brackets (or no brackets at all) (i.e obcDriverConnect( should be obcDriverConnect() or obcDriverConnect)

b. Page 15 look says "install. Packages" should be "install.packages"

c. Page 22 "page gives the complete programming language" should read "page gives the complete programming code"

d. Page 25 HTLM -> HTML

Response:

a. We went through all the brackets and closed any open ones.

b. We changed it to “install.packages”.

c. We changed “language” to “code”.

d. We changed “HTLM” to “HTML”

Reviewer 3

1) What process did the authors use to select these variables and not others to appear in the dashboards? There are many data points left out.

Response: 

The purpose of this paper was to demonstrate the development of this dashboard. The variables chosen for inclusion in the dashboard are meant to inform higher education administrators, institutional researchers, and postsecondary policymakers. However, this paper is intended to inform the reader how a similar dashboard using large, publicly available datasets might be created. We encourage the readers of this paper to study all the data sources and dictionaries of any dataset they are interested in using. To address this, we added this language to the paper:

” These datasets contain thousands of variables. IPEDS has up to 250 variables [4], the USCB over 18,000 [3] and BLS has several datasets containing tens of thousands of variables (would could find no final number) [5]. For this demonstration, we choose variables related to demographics, educational outcomes and labor outcomes. However, other programmers may choose to use any number of variables from these datasets depending on the goals of their project. No matter what variables are chosen, it is imperative that the programmer studies the variable sources, code books, and logic. This is the case whether they choose these demonstrated datasets, or other datasets. It is our goal to use the variables in this paper to demonstrate basic skills that could be applied to any number of variables from any number of data sources”

2) A main claim in the paper is that there isn’t a dashboard cited in the literature that combines US federal census, labor and education. Are there any studies that combine this data themselves without a dashboard? This would strengthen the argument that this dashboard was needed and provide a basis to evaluate the effectiveness of this dashboard in future years.

Response:

On our first submission, we had some literature review on the few sources that we found that combine these types of data. We have re-introduced a short paragraph about those as we agree it adds to the context or need. You will note a new section titled “These Data Sets Combined”. 

3) Some discussion is needed about the logistics of hosting and maintaining this dashboard. The authors seem to be using a free account on shinyapps.io. What are the limitations as the dashboard is used more frequently for this approach? Did the authors consider other deployment paths? (i.e. was cost the only factor, or were other deployment options evaluated).

Response:

We have added a section on deployment and hosting of the dashboard called “Deployment of the Dashboard”. We note that the free version of Shiny comes with a data limit and that there are purchasable subscriptions if more data needs to be stored. We also note that this demonstration allows for the creation and deployment of a Shiny application. R does offer other applications where this code may be useful, but the focus of this project was to use Shiny since it is free and open-source.

4) What plans are there to maintain the data set and what is the workload? i.e. what effort is needed to add 2020 data and beyond? How long will that take given the initial code is in place?

Response:

We have added a section at the end called “Maintenance and Updates” where we discuss the need to update the data each year. We provide a schedule of when Census, BLS, and IPEDS data are updated and suggest updating all of the simultaneous with IPEDS data. We note that updating the data requires an adjustment of term codes in the programming language and the obtainment of the most recent data sets from each entity. We also note that with practice, updating the data need not take more than a few hours a year.

5) The response to a prior reviewer comment about 2-year or 4-year institutions I think misses the mark. This is a somewhat misleading variable in IPEDS that actually can lead to wrong conclusions if a researcher is unaware what it represents. IPEDs codes institutions as 4-year institutions if they have any bachelor’s degree programs. 2-year institutions (in the IPEDS data set) are those institutions with only associates degrees. Most times this is not actually what a researcher actually wants and excludes nearly all community & technical colleges in some states that award primarily associates degrees and less than two year certificates, but have a few bachelor’s degree programs.

Response: 

We agree with you and have added language that encourages the user, if wanting to use this variable, to study the IPEDS data dictionary on the ACCESS file. Under the “valuesetsYY” tab, you can look up the CYYSZSET variable in the HDYYYY table and note the different categories as they are coded by IPEDS. These include a range of categories from “Two-year, very small” for a coded value of 1, to a “Not applicable” category for a value of 20. We thus agree that this is a misleading variable if the researcher wants to use IPEDS data for other purposes and have added language in our paper to emphasize that the researcher or coder should consider the limitations of the variable names in IPEDS or in any other source, but for purposes of demonstrating how to reprogram variables in R, we have used this variable.

You will also note that we included our coding logic for that variable in the paper, which is followed by this statement:

“It is important to consider the objectives of a dashboard when conducting research with IPEDS or other data. For example, this particular dataset may not include every category of higher education institutions. For example, some institutions that are not public, only award less than two year certificates, or other types may not be classified in the desired way by IPEDS or other data sources. Therefore, it is imperative of the researcher to choose datasets that meet their dashboard’s objectives.”

Thank you once again for your feedback.

Sincerely,

Authors

---

## [Decision Letter · Decision Letter 2]

21 Nov 2022

Using R Shiny to Develop a Dashboard using IPEDS, U.S. Census, and Bureau of Labor Statistics Data

PONE-D-21-37932R2

Dear Dr. Perkins,

We’re pleased to inform you that your manuscript has been judged scientifically suitable for publication and will be formally accepted for publication once it meets all outstanding technical requirements.

Kind regards,

Sathish A.P. Kumar

Academic Editor

PLOS ONE

Additional Editor Comments (optional):

Reviewers' comments:

Reviewer's Responses to Questions

**Comments to the Author**

1. If the authors have adequately addressed your comments raised in a previous round of review and you feel that this manuscript is now acceptable for publication, you may indicate that here to bypass the “Comments to the Author” section, enter your conflict of interest statement in the “Confidential to Editor” section, and submit your "Accept" recommendation.

Reviewer #3: (No Response)

2. Is the manuscript technically sound, and do the data support the conclusions?

Reviewer #3: Yes

3. Has the statistical analysis been performed appropriately and rigorously? 

Reviewer #3: N/A

4. Have the authors made all data underlying the findings in their manuscript fully available?

Reviewer #3: Yes

5. Is the manuscript presented in an intelligible fashion and written in standard English?

Reviewer #3: Yes

6. Review Comments to the Author

Reviewer #3: The authors have addressed my feedback in the prior review. I have no other feedback for this paper.

7. PLOS authors have the option to publish the peer review history of their article (what does this mean?). If published, this will include your full peer review and any attached files.

Reviewer #3: No

---

## [Editor Report · Acceptance letter]

25 Nov 2022

PONE-D-21-37932R2 

Using R Shiny to Develop a Dashboard using IPEDS, U.S. Census, and Bureau of Labor Statistics Data 

Dear Dr. Perkins:

I'm pleased to inform you that your manuscript has been deemed suitable for publication in PLOS ONE. Congratulations! Your manuscript is now with our production department. 

Kind regards, 

on behalf of

Dr. Sathish A.P. Kumar 

Academic Editor

PLOS ONE